# What techniques might be used to harness placebo effects in non-malignant pain? A literature review and survey to develop a taxonomy

Felicity L Bishop,[1] Beverly Coghlan,[2] Adam WA Geraghty,[2] Hazel Everitt,[2] Paul Little,[2] Michelle M Holmes,[1] Dionysis Seretis,[1] George Lewith[2]

[1]Department of Psychology, Faculty of Social Human and Mathematical Sciences, University of Southampton, Southampton, UK
[2]Primary Care and Population Sciences, Aldermoor Health Centre, University of Southampton, Southampton, UK

**Correspondence to**
Dr Felicity L Bishop; F.L.Bishop@southampton.ac.uk

## ABSTRACT

**Objectives** Placebo effects can be clinically meaningful but are seldom fully exploited in clinical practice. This review aimed to facilitate translational research by producing a taxonomy of techniques that could augment placebo analgesia in clinical practice.

**Design** Literature review and survey.

**Methods** We systematically analysed methods which could plausibly be used to elicit placebo effects in 169 clinical and laboratory-based studies involving non-malignant pain, drawn from seven systematic reviews. In a validation exercise, we surveyed 33 leading placebo researchers (mean 12 years' research experience, SD 9.8), who were asked to comment on and add to the draft taxonomy derived from the literature.

**Results** The final taxonomy defines 30 procedures that may contribute to placebo effects in clinical and experimental research, proposes 60 possible clinical applications and classifies procedures into five domains: the patient's characteristics and belief (5 procedures and 11 clinical applications), the practitioner's characteristics and beliefs (2 procedures and 4 clinical applications), the healthcare setting (8 procedures and 13 clinical applications), treatment characteristics (8 procedures and 14 clinical applications) and the patient–practitioner interaction (7 procedures and 18 clinical applications).

**Conclusion** The taxonomy provides a preliminary and novel tool with potential to guide translational research aiming to harness placebo effects for patient benefit in practice.

## Strengths and limitations of this study

► This is a novel attempt to use existing studies to identify the factors that might contribute to placebo effects and the associated procedures that could be simply and ethically adapted for clinical practice, subject to further testing.
► We drew on both clinical trials and laboratory-based studies of placebo effects, in order to generate a more comprehensive list of factors that might contribute to placebo effects than would be possible by relying on just one literature.
► A systematic approach to data synthesis, based on qualitative research methods, was used to identify and classify procedures that might contribute to placebo effects in clinical trials.
► The development of the taxonomy did not incorporate very recent placebo trials or studies and the selection of reviews used to determine which original studies to include in the development process was somewhat arbitrary.
► Our taxonomy is presented not as an exhaustive compilation of current methods used in placebo research but as a detailed and systematic guide for future research, which can in turn further refine the taxonomy.

## INTRODUCTION

There is compelling evidence that factors other than the so-called active components of treatment can have clinically meaningful effects on symptoms, particularly non-malignant pain.[1–4] Such 'placebo effects' can be defined as the physiological and/or psychological changes that result from the meaning derived by a person in a healthcare setting.[5 6] Expectations—which can be generated, for example, by verbal suggestion or previous experience—play a key role in placebo effects.[7] These effects may be as large as treatment effects[8] and occur throughout medicine, especially when doctors and patients interact with each other. They are not routinely deliberately harnessed for patient benefit in clinical practice,[9] possibly because doctors often assume they must deceive patients in order to elicit placebo effects.[10 11] However, this assumption is mistaken because it is not necessary to prescribe placebos in order to elicit placebo effects. For example, the overall analgesic effect of an opioid derives from its specific pharmaceutical actions and its psychological components, that is, the expectations and meaning that the patient derives when consulting the doctor and taking the

medicine.[12 13] The same is true for other types of intervention including physical, surgical and psychotherapies. One approach that has received initial support is for doctors to use positive suggestion to enhance patients' expectations of benefit.[4] Furthermore, preliminary evidence suggests that openly prescribing placebos might elicit clinically meaningful placebo effects in irritable bowel syndrome (IBS) and depression,[14 15] although this approach entails its own set of ethical challenges.[16 17]

Placebo researchers have called for more translational research in this field.[11 18–20] Such work has thus far typically focused on ethical considerations and narrative approaches to drawing out implications for clinical practice from the placebo literature. We suggest a systematic approach to translational research might be helpful. Many techniques or procedures contribute to placebo effects and could potentially be simply and ethically adapted for clinical practice, subject to further testing in practice settings.[21] In order to identify and describe such techniques and thus provide some direction for future research, we reviewed experimental and clinical studies of placebo effects in non-malignant pain. We focused on non-malignant pain because it can be difficult to manage (particularly with current concerns about opioids[22]), the mechanisms underpinning placebo analgesia are reasonably well understood,[23] laboratory-based experimental studies often focus on placebo analgesia and patients with pain have been shown to display substantial and clinically significant placebo effects.[1] The aim of this project was to facilitate translational research by producing a taxonomy of techniques that may contribute to placebo effects observed in research settings and could be studied as options for augmenting placebo enhancement of analgesia in clinical practice.

## METHODS
### Literature search
We selected seven systematic reviews of different aspects of the placebo literature, chosen from recent reviews available at the time (2012) and based on expert opinion (within the research team) to enable the extraction of information on placebo procedures from a broad range of settings—comprehensive reviews,[24–26] reviews of placebo effects in clinical populations[2 27] and reviews of laboratory-based experimental placebo studies.[28 29] The key consideration was that this collection of reviews should cite a diverse set of studies likely to be using diverse methods to directly (eg, placebo mechanisms studies) or indirectly (eg, clinical trials with placebo controls) study placebo effects. After removing duplicates and ineligible studies (see figure 1), 169 studies were used to develop the taxonomy (for a list of included studies, see online supplementary material).

Studies were eligible for inclusion if they reported original research in which some participants received a placebo intervention, reported a non-malignant pain outcome, were published since 1983 and were published in English language. Studies were excluded if they were published before 1983 (because (A) means of generating

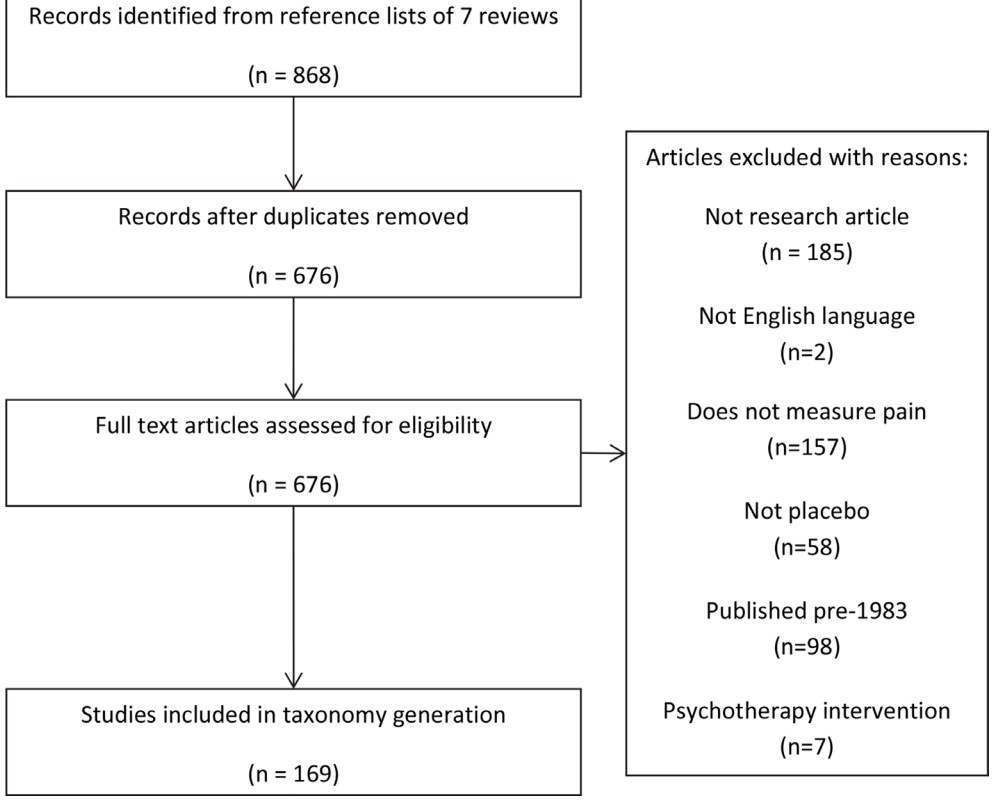

**Figure 1** Flowchart showing identification of studies.

context-dependent placebo effects may be sensitive to social and cultural changes over time, eg, patient preferences for particular communication styles and thus their effectiveness in modifying expectations may have changed over time and (B) this yielded a manageable number of papers to analyse which had been published during the 30 years preceding this analysis) or examined any type of psychotherapeutic interventions (because it is difficult to disentangle the active components of psychotherapy from the effect of the meaning of the intervention[30]).

## Data extraction and synthesis

Descriptions of all events that occurred in the placebo groups during each of the 169 studies (eg, medical, administrative and ethical procedures) were extracted into a piloted form by one author and checked by a second. These events were reviewed for duplication and overlap. This resulted in an initial list of 43 procedures that might contribute to placebo effects (eg, informed consent processes, taking placebo pills, conditioning protocols). Study authors were not contacted for further information.

To synthesise the data and develop our taxonomy, we used systematic and rigorous methods derived from qualitative research. We began with a deductive analysis, which aimed to categorise the procedures in a way that is intuitively appealing, accessible and clinically relevant by sorting them into five previously identified contextual domains of healthcare: patients' characteristics/beliefs, practitioners' characteristics/beliefs, patient-practitioner interaction, superficial treatment characteristics and the healthcare environment.[31] Two authors (BC and FLB) performed the initial categorisation which was then reviewed in detail by three other authors (GL, HE and AG). We then engaged in a constant comparative analysis, a technique that originates in grounded theory.[32] The aim of this part of the analysis was to consolidate the list of procedures and ensure that we only retained those that were distinct from each other. Procedures and examples of their use were all systematically compared with each other; similar procedures were then merged, and all procedures were classified into one of the five domains. Two authors (BC and FLB) led this work and presented initial findings to the rest of the team for discussion. All authors discussed and agreed on which procedures to merge, which to retain and how to classify them. During this process, the definitions of the five domains were iteratively modified in order to reduce ambiguity over which procedures should be classified into which domain. This resulted in a more parsimonious list of 29 procedures classified by domain. All authors discussed and agreed on the final classification of these procedures. These 29 procedures were then critically examined to ensure they were theoretically plausible means of producing placebo effects. We focused on three core psychological mechanisms[25] [33–36]: response expectancy[37]; conditioning and social learning[7] and affect, including motivation and anxiety reduction.[24] [38] However, we acknowledge that

these mechanisms are difficult to tease apart[39] and that alternative mechanisms have been proposed[6] and so we erred on the side of inclusivity. Neurobiological mechanisms of placebo analgesia have been described,[23] [40] but a detailed consideration of how these might apply to the procedures in the taxonomy would be highly speculative and was beyond the scope of this project (for discussion of clinical applications of the neuroscience of placebo effects, see Jubb and Bensing[41]). Four authors (FLB, BC, AG and GL) reviewed all procedures and considered the extent to which each procedure could plausibly produce placebo effects via one or more of the three core psychological mechanisms. Initial findings were shared with the remaining authors, and consensus was reached through discussion. Four procedures deemed very unlikely to produce placebo effects (conveying a neutral therapeutic message, randomisation, blinding, deception) were excluded, leaving 25 procedures that might plausibly contribute to placebo effects. The multidisciplinary team of authors (including general practitioners (GPs), clinical and health psychologists and complementary medicine specialists, for example) then generated possible clinical applications of each of these 25 procedures.

## Validating the taxonomy

To ensure our taxonomy was comprehensive, we surveyed leading placebo researchers (authors of major publications on placebo effects, attendees at an international symposium on placebo effects and GPs with an interest in placebo effects). These researchers were identified from the systematic reviews and references used to develop the taxonomy; the list of attendees at Beyond The Placebo: Biomedical Clinical and Philosophical Aspects of the Placebo Effect, held in Ascona Switzerland, August 2012; and GPs within the National Institute for Health Research School for Primary Care Research. Ethical approval for the survey was obtained from the host institution (reference: 4741). Completed electronic surveys including informed consent were received from 33 researchers (52% response rate) experienced in placebo research (mean 12 years' experience, SD 9.8). Respondents were shown our draft taxonomy and asked whether, for each domain, they knew of any other procedures that could elicit placebo responses. The proportion answering yes ranged from 22% (healthcare setting domain) to 50% (superficial treatment characteristics domain). Respondents suggested 85 additions which were screened against existing procedures and for theoretical plausibility: 80 of the suggested additions were extra details or suggested clinical applications of existing procedures; five were new and distinct plausible procedures that were added to the taxonomy, giving a final total of 30 procedures.

Because of our orientation to clinical applications, we have chosen to use clinically oriented terminology throughout the taxonomy. However, it is important to note that when used in relation to procedures identified from the literature, these terms also relate to the experimental equivalent, such that 'patient' also refers

to subject/participant, 'practitioner' also refers to experimenter and 'intervention' also refers to experimental condition.

## Analysis

The use of each of the 30 procedures in the taxonomy was assessed across all 169 studies in the review. Two authors independently rated the presence of each procedure in each study (kappa 0.93, discrepancies were resolved through discussion).

## RESULTS

The taxonomy defines 30 procedures that may contribute to placebo effects observed in clinical and experimental research and classifies them into five domains. Table 1 presents the main taxonomy, listing and defining all 30 procedures within five domains. Table 2 suggests clinical applications of each procedure. Table 3 shows the frequency of use of each procedure in clinical and experimental studies and is intended as both an approximate guide to whether the procedures derived primarily from one or other literature and as a means to highlight those procedures that are very common and very rare in the literature. Below we describe the procedures within each domain in turn.

### Patient's characteristics and beliefs

The taxonomy specifies five procedures that act directly on the patient's characteristics and/or beliefs in ways that might contribute to placebo effects. Procedure 1 involves selecting patients who are most likely to benefit from an intervention based on their history of similar treatments (where similarity is construed broadly at multiple levels, including appearance, modality, style and pharmacology). For example, one might select those patients who have not experienced disappointing results from a similar intervention in the past (as the latter group might have learnt to expect the intervention to fail). This procedure was commonly used by clinical trials and (to a lesser degree) experimental studies.

Procedures 2 (create positive expectancy), 3 (reduce negative expectancy) and 4 (convey a positive therapeutic message) all involve communicating with patients to encourage them to expect beneficial effects of treatment or not to expect side effects. The majority of experimental studies in our review explicitly encouraged patients to expect treatment benefits, while very few clinical studies explicitly targeted patients' expectations and hardly any studies attempted to minimise patients' expectations of side effects. Procedure 5 involves tailoring the intervention to the patient's sociocultural context. This approach emerged from the expert feedback, and while it seems plausible and ethical to translate into clinical practice, it was not used by any of the reviewed studies.

The procedures in the patients' beliefs and characteristics domain are thought to contribute to placebo effects primarily through altering patients' response expectancy. Selecting patients based on treatment history and tailoring

to sociocultural context are also predicated on learning mechanisms, that is, learnt associations between treatment outcome and treatment properties. There is some evidence that clinicians can give verbal suggestions to alter patients' expectations in practice and that this reduces patients' pain, particularly acute procedural pain.[4 42] As part of work to implement these procedures more widely in practice, it would be important to investigate how to secure ethically valid consent for treatment. For example, clinicians might want to encourage realistically positive patient expectations while providing information about possible harms without inducing the negative expectations that could trigger nocebo effects.[43 44]

### Practitioner's characteristics and beliefs

The two procedures in this domain are about using or modifying healthcare practitioners' characteristics and/ or beliefs. Procedure 6 requires a practitioner to expect a treatment to benefit the patient. This might contribute to observed placebo effects in patients by influencing a practitioner's communication about the treatment and hence a patient's response expectations and/or affective response to the consultation. Only 1% of clinical studies and no experimental studies reported modifying practitioners' expectations. This procedure has received little attention in the placebo literature, but clinical research in musculoskeletal settings suggests that practitioners' outcome expectations can predict patients' pain outcomes.[45] One way to implement this procedure in practice would be for practitioners to communicate explicitly that they believe a treatment is effective, an approach which clearly overlaps with communication interventions designed to help doctors encourage patients to have positive expectations. Implementing procedure 6 also depends on practitioners having relevant high-quality evidence readily available and accessible and understanding this evidence as it applies to the patient.

A small proportion of studies (9% of experimental studies and no clinical studies) emphasised a practitioner's status or other characteristics (procedure 7). For some patients, a high-status practitioner might elicit more confidence in the treatment (and thus higher expectations) and/or a more positive affective response to the consultation.[46] Some aspects of this procedure are already part of clinical practice, for example, the routine display of medical certificates in doctors' offices; others are inherent in the tools of the doctor, such as the symbolic properties of the stethoscope.[47] However, there is likely to be scope for testing their effects and augmenting their use if appropriate.

### Healthcare setting

Procedures 8 and 9 relate to the efforts made in studies to actively recruit and retain patients, respectively. Clinical and experimental studies both reportedly used these procedures sparingly (<20% for active recruitment and <5% for active retention). Such efforts may make patients feel valued and could be implemented

**Table 1** Taxonomy of procedures which could plausibly elicit placebo effects in non-malignant pain

| Procedure derived from literature | Definition and use in research studies |
|---|---|
| **Patient's beliefs and characteristics** | |
| 1. Select patients based on treatment history | Screen and select patients (or subgroups) against inclusion criteria related to issues such as medical/treatment history, for example, naive to intervention being tested (not just contraindications). |
| 2. Create positive expectancy | Deliberately and explicitly suggest to patients that the intervention will be effective for them (not as part of informed consent process). |
| 3. Reduce negative expectancy* | The potentially negative or harmful procedures and characteristics of the treatment are deliberately minimised in information for patients. |
| 4. Convey a positive therapeutic message through informed consent procedures | Convey (verbally or in writing) a positive therapeutic message through the content of informed consent. The message might be explicit (eg, 'this intervention is usually effective in most people') or implicit (eg, 'this treatment is an antihypertensive'). |
| 5. Harness sociocultural context* | Tailor the intervention according to the patient's social and cultural context and history. |
| **Practitioner's beliefs and characteristics** | |
| 6. Practitioner expectancy | The person delivering the treatment expects it to be effective for the patient. |
| 7. Practitioner's personal characteristics | The practitioner's personal and/or professional characteristics (eg, status) are modified (through selecting practitioners with different characteristics) and/or emphasised to patients. |
| **Healthcare setting** | |
| 8. Active recruitment | Actively seek out and recruit patients (eg, advertising for specific types of patients, writing personally to individual eligible patients identified through medical records). |
| 9. Active retention | Make patients feel valued by attempting to keep them in a study (eg, contact patients if they miss an appointment, incentivise attendance through monetary or non-monetary gifts). |
| 10. Follow-up | Assess patients after the intervention/experiment to assess long-term maintenance or changes in effects over at least 6 months. |
| 11. Follow a standardised protocol | The intervention is delivered according to a set, scientifically derived protocol, lending credibility to the intervention (and is therefore not individualised for each patient). |
| 12. Ethical oversight | Study practices and procedures are explicitly regulated and monitored by an institutional ethics committee, lending credibility to the intervention. |
| 13. Participating in research | Patients know that they are part of research and contributing to the furthering of human knowledge and/or improvement of healthcare for future patients. |
| 14. Symptom monitoring | Monitor patients' symptoms using self-report measures, practitioner assessment or objective measures repeated over time at least twice; patients are aware of the resulting measurements. |
| 15. Enhanced environment* | The physical and interpersonal environment where the intervention is delivered is deliberately enhanced. |
| **Treatment characteristics** | |
| 16. Sham intervention— medication | An inert substance is administered which is manufactured to appear identical to an active medication (eg, sugar pill, intravenous saline, topical agent). |
| 17. Sham interventions— physical | A sham physical intervention is administered which is designed to appear identical to the genuine intervention (eg, deactivated transcutaneous electrical nerve stimulation (TENS), non-penetrative acupuncture needles at non-acupuncture points). |
| 18. Sham interventions— attention only | Patients receive study-specific attention in terms of numbers of visits and time spent with study staff but no additional intervention. |
| 19. Ineffective substances* | Products unlikely to be effective or not indicated are administered (eg, vitamins in the absence of vitamin deficiency). |
| 20. Use salient side effects | Potential side effects are highlighted such that the patient can interpret them as evidence of a potent intervention. |
| 21. Matched treatments | To secure blinding, placebo/sham treatments are matched to 'real' treatments (eg, on mode of administration, dosage, frequency of administration, visual appearance, taste, smell, individual titration procedures). |

**Table 1** Continued

| Procedure derived from literature | Definition and use in research studies |
|---|---|
| 22. Maximised treatment procedures | The procedures and characteristics of the treatment are exaggerated, for example, through high dose, use of colour, high frequency, large pill size, lengthy duration of intervention, ritualistic administration. |
| 23. Conditioning | A desired response (eg, pain relief) is paired with an intervention stimulus (eg, placebo cream) so that the patient associates the response with the stimulus. |
| **Patient–practitioner interaction** | |
| 24. The process of informed consent | The patient's formal written and/or verbal informed consent is discussed and obtained. |
| 25. Detailed history | A detailed personal and/or medical and/or psychosocial history is obtained from the patient. |
| 26. Diagnosis/tests | Additional tests, examinations or confirmatory diagnostic procedures are undertaken to establish eligibility for the study. |
| 27. Care | The practitioner deliberately engages the patient with warmth, compassion and empathy. |
| 28. Patient-centred communication* | The practitioner adopts a style of consultation that they consider to be appropriate for a particular patient. |
| 29. Extra attention | The patient receives extra attention from being in the study, for example, is seen more frequently or for longer than usual. |
| 30. Continuity of care | Efforts are made for the same practitioner to see the same patient at each contact. |

*Procedures added following survey of researchers.

in practice through the use of personalised communications from practitioners to encourage attendance at appointments.

Three of the eight procedures in this domain were used by over half of clinical and experimental studies and relate to basic structural features of research: following a protocol, ethical oversight and participating in research (procedures 11–13). They are thought to impact patients' expectations, by emphasising the legitimacy of the intervention that is being provided and the importance of the patient's contribution to a bigger project, that is, generating knowledge. Translating these procedures into practice could involve, for example, clinicians explicitly talking with patients about official guidance and treatment protocols that they are following.

Symptom monitoring (procedure 14) was commonly used in both clinical and experimental studies. This could be implemented in practice, for example, through repeatedly using patient-reported outcome measures (see Snyder et al[48]) and might contribute to placebo effects through learning mechanisms (eg, regular symptom monitoring acts as feedback to motivate health behaviours and/or modify patients' goals). Alternatively, the mere act of asking a patient to monitor their symptoms could convey an expectation of treatment benefit, altering the meaning of a clinical interaction for the patient. Traditionally, such effects of the act of measurement are dismissed as Hawthorne effects, but they may also be encompassed in broader definitions of placebo effects as meaning effects[49] and could thus enhance effects in clinical practice despite being considered a nuisance in clinical research.

Very few placebo studies (5% of experimental and no clinical studies) reported enhancing the physical or interpersonal environment (procedure 15). There is a separate and distinct literature on environment modifications in health settings that might be usefully integrated with the placebo literature when developing clinical applications in this area and modelling mechanisms of action.[50 51]

### Treatment characteristics

Eight procedures in the taxonomy involve modifying the characteristics of a treatment. Three involve prescribing sham interventions (sham medication, procedure 16; sham physical interventions, procedure 17 and extra attention, procedure 18), while a fourth involves prescribing a substance unlikely to be effective for the symptom in question (procedure 19). These four procedures represent variations in control conditions used in research and were frequently used by both clinical and experimental studies (with the exception of extra attention which was only used by 2%–5% of studies). Such controls are thought to operate primarily via expectations, while affective pathways may also be important when extra attention from trial personnel/medical staff is involved. Of all the procedures in the taxonomy, these four that represent control conditions come closest to the traditional notion of how placebos could be applied in practice. Given ethical concerns around deceptive prescribing, we suggest that translational research might continue to focus on openly prescribing sham interventions including placebo pills (as in Refs14 and 15). Other options should not be

**Table 2** Suggested potential clinical applications of procedures to elicit placebo effects in non-malignant pain, subject to further research

| | Procedure | Suggested clinical applications |
|---|---|---|
| **Patient's beliefs and characteristics** | | |
| 1. | Select patients based on treatment history | Stop prescribing interventions of a type that a patient has previously not responded to (eg, tablets); instead, prescribe a different, new type of treatment (eg, psychological therapy). |
| 2. | Create positive expectancy | Tell the patient the intervention is likely to be effective. Elicit patients' treatment and illness beliefs and expectations and dispel any misconceptions. Empower patients to self-care. |
| 3. | Reduce negative expectancy | Limit emphasis on major potential side effects and describe how uncommon they are. Hide cessation of analgesia administration (eg, as in Benedetti et al[73]), after obtaining advanced consent and ensuring patients are aware they can request additional analgesia if needed. |
| 4. | Convey a positive therapeutic message through informed consent procedures | Provide written and/or verbal information that conveys a positive therapeutic message about treatment. Provide clear rationale for treatment. Provide patient testimonials and supporting literature/media. |
| 5. | Harness sociocultural context | Elicit patients' culturally embedded treatment and illness beliefs, preferences and expectations, dispelling any potentially harmful misconceptions. Involve significant others in care. |
| **Practitioner's beliefs and characteristics** | | |
| 6. | Practitioner expectancy | Only prescribe a treatment to patients when the practitioner expects it will be effective; communicate that expectation to patients. |
| 7. | Practitioner's personal characteristics | Honour patient preferences for particular practitioners. Use indicators of expertise/high status in offices, in correspondence and when referring to other practitioners. Ensure the patient is seen by a practitioner whose views/values are congruent with the patient's views/values. |
| **Healthcare setting** | | |
| 8. | Active recruitment | Actively seek out patients and invite them to attend clinic regarding a particular intervention (as opposed to waiting for patients to present). |
| 9. | Active retention | Personally contact patients if they miss an appointment. Use incentives to encourage patients to keep appointments. |
| 10. | Follow-up | Routinely invite patients to book a follow-up appointment after an intervention has finished and prior to repeat prescription. Encourage the patient to take responsibility for and self-manage their condition following an intervention. |
| 11. | Follow a standardised protocol | Use patient-friendly treatment protocols and share with patients where they fit in that protocol. |
| 12. | Ethical oversight | Ensure that patients understand that their treatment protocol is sanctioned by a higher authority, for example, National Institute for Health and Care Excellence. |
| 13. | Participating in research | Inform patients that all outcomes and practitioner performance is audited and can contribute to improved knowledge and treatment for future patients. |
| 14. | Symptom monitoring | Ask patients to monitor their symptoms regularly, for example using email, phone apps, web-based systems, paper forms. Assess treatment outcome. Give patients feedback on symptom improvements following monitoring. |
| 15. | Enhanced environment | Ensure that the environment is professional, pleasant and peaceful. Employ friendly and helpful support staff. |
| **Treatment characteristics** | | |
| 16. | Sham intervention—medication | Openly prescribe sham medication. With advanced prior consent, prescribe sham medication. |

**Table 2**  Continued

| | Procedure | Suggested clinical applications |
|---|---|---|
| 17. | Sham interventions—physical | Openly prescribe sham physical treatments. |
| | | With advanced prior consent, prescribe sham physical treatments. |
| 18. | Sham interventions—attention only | Increase frequency and duration of consultations. |
| 19. | Ineffective substances | Prescribe substances that are likely not to cause harm but not clearly indicated or substances unlikely to be effective, for example, simple linctus. |
| 20. | Use side effects | Tell patients about side effects associated with positive clinical outcome. |
| 21. | Matched treatments | Design appearance of prescribed substance (eg, colour, packaging, taste) to match known effective treatments. |
| 22. | Maximised treatment procedures | Within safety limits prescribe higher dose/higher frequency/larger pill. |
| | | Use different colour treatments. |
| | | Instigate ritualistic procedures patients can perform when taking medicines. |
| | | Maximise adherence to treatment through education, easy follow-up appointments, easy repeat prescription arrangements, and so on. |
| 23. | Conditioning | Prescribe highest tolerated dose first, then titrate downwards. |
| | | With consent, begin with active intervention, pair with a seemingly identical placebo then substitute for placebo alone (eg, as in Sandler and Bodfish[56]). |
| **Patient–practitioner interaction** | | |
| 24. | **The process of informed consent** | Actively seek patient consent. |
| | | Provide treatment options and encourage the patient to choose from these options if they so desire. |
| 25. | Detailed history | Take a detailed medical and psychosocial history/update. |
| | | Ensure the patient feels listened to, for example, through non-verbal communication and/or capturing information. |
| | | Ask questions about the meaning of symptoms. |
| 26. | Diagnosis/tests | Provide a definitive/confident diagnosis. |
| | | Examine the patient fully. |
| 27. | Care | Allow patient adequate time to tell their story and listen to them. |
| | | Validate the patient's concerns. |
| | | Use non-verbal techniques to convey empathy, compassion, warmth. |
| | | Use touch judiciously. |
| 28. | Patient-centred communication | Individualise consultation style according to a patient's preference for example, collaborative versus authoritative. |
| | | Engage in collaborative decision-making with the patient. |
| | | Develop shared treatment goals that you and the patient agree on. |
| 29. | Extra attention | Give extra attention to or show more interest in a patient by seeing them more frequently, having longer consultations or visiting at home. |
| | | Do not rush the patient. |
| 30. | Continuity of care | Ensure patient is cared for by the same practitioner. |
| | | Read records before consultation. |

Suggestions for clinical applications pending research into effectiveness and ethical acceptability in clinical settings.

dismissed entirely though: advanced consent and even waiving consent are acceptable to some patients, and so, it is vital for translational research to continue exploring patients', practitioners' and other stakeholders' views on the acceptability and ethics of diverse ways of prescribing placebos.[52–55]

Three procedures in this domain modify the superficial (non-pharmacological or non-defining) characteristics of treatments. Procedure 20 is to highlight treatment side effects to patients in order to encourage patients to see the treatment as potent; this procedure was very rare, used by only 1% of clinical studies. Procedure 21 was much more commonly used and involves matching the

**Table 3** Use of procedures in placebo groups of clinical and experimental studies

| Procedure | % of studies that used each procedure | |
|---|---|---|
| | Experimental (n=58) | Clinical (n=111) |
| **Patient's beliefs and characteristics** | | |
| 1. Select intervention based on patient's treatment history | 55 | 75 |
| 2. Create positive expectancy | 76 | 5 |
| 3. Reduce negative expectancy | 3 | 0 |
| 4. Convey a positive therapeutic message through informed consent procedures | 43 | 1 |
| 5. Harness sociocultural context | 0 | 0 |
| **Practitioner's beliefs and characteristics** | | |
| 6. Practitioner expectancy | 0 | 1 |
| 7. Practitioner's personal characteristics | 9 | 0 |
| **Healthcare setting** | | |
| 8. Active recruitment | 14 | 16 |
| 9. Active retention | 3 | 2 |
| 10. Follow-up | 2 | 16 |
| 11. Follow a standardised protocol | 85 | 63 |
| 12. Ethical oversight | 78 | 69 |
| 13. Participating in research | 86 | 84 |
| 14. Symptom monitoring | 95 | 89 |
| 15. Enhanced environment | 5 | 0 |
| **Treatment characteristics** | | |
| 16. Sham intervention—medication | 71 | 55 |
| 17. Sham interventions—physical | 33 | 41 |
| 18. Sham interventions—attention only | 2 | 5 |
| 19. Ineffective substances | 0 | 1 |
| 20. Use side effects | 0 | 1 |
| 21. Matched treatments | 40 | 82 |
| 22. Maximised treatment procedures | 22 | 3 |
| 23. Conditioning | 41 | 0 |
| **Patient–practitioner interaction** | | |
| 24. The process of informed consent | 88 | 77 |
| 25. Detailed history | 19 | 33 |
| 26. Diagnosis/tests | 36 | 41 |
| 27. Care | 0 | 1 |
| 28. Patient-centred communication | 0 | 0 |
| 29. Extra attention | 2 | 63 |
| 30. Continuity of care | 7 | 14 |

appearance of real and control treatments (used by 40% of experimental and 82% of clinical studies), in order to maintain patient blinding. This could be translated into clinical practice by designing the appearance of interventions to match patients' beliefs about what effective interventions look like. Procedure 22 involves maximising or exaggerating the superficial characteristics of treatment in order to generate larger placebo effects, for example, by using colour, large pill size or ritualistic administration of medicines, manipulations which could alter the meaning of a treatment for a patient and/ or enhance their expectations. Twenty-two per cent of experimental studies reported using this procedure, and one way to translate it into practice would be to create (and test) ritualistic procedures for patients to engage in when taking medicines.

The final procedure in this domain—procedure 23, conditioning to generate placebo effects—was used

commonly and exclusively by experimental studies (41%). Conditioning protocols generate placebo effects through learning mechanisms and perhaps could be implemented in practice to reduce pharmaceutical dosages, as was achieved in a pilot study in children with attention-deficit disorder.[56]

### Patient–practitioner interaction

The patient–practitioner interaction domain incorporates seven procedures related to the interpersonal relationship or interactions between a patient and their healthcare practitioner. These procedures are thought to operate primarily through affective mechanisms such as reduced anxiety after telling one's story and being listened to with empathy and acknowledged, although more cognitive pathways via expectations are also plausible.[57] Three procedures are about specific processes that can occur during consultations—obtaining informed consent (procedure 24), taking a detailed history (procedure 25) and performing additional diagnoses or tests (procedure 26). Arguably, these procedures indicate to the patient that the practitioner respects them, is interested in their perspective and is thorough in their diagnosis. They occur in both clinical and experimental research settings and could be relatively directly translated into practice or optimised if already used.

Two procedures are about the way in which the practitioner engages with the patient: communicating care (procedure 27) and patient-centred communication (procedure 28). These procedures were surprisingly very rarely described in the studies included in our review, although recently, the nocebo effects of not validating a patient's experiences have been shown to be particularly potent.[58] There is of course a distinct and large literature on doctor–patient communication, and fruitful dialogue is beginning to bridge these fields.[59]

The final two procedures in this domain refer to more structural aspects of consultations: extra attention (procedure 29, ie, longer or more frequent appointments) and continuity of care (procedure 30). Sixty-three per cent of clinical studies used extra attention, while a small proportion of clinical (14%) and experimental (7%) studies reported providing continuity of care. Directly implementing these procedures in practice might be challenging given the ever-increasing constraints on healthcare resources and drives to reduce cost.

### DISCUSSION

The taxonomy names and describes 30 procedures that may contribute to placebo effects in experimental and clinical studies and classifies them into five domains. It includes 60 theoretically plausible clinical applications, subject to further research on their effectiveness and ethical acceptability in practice. Some of the clinical applications derived from the placebo literature have already been investigated in their own right under other auspices, highlighting the need for the burgeoning translational

science of placebo effects to be broad ranging and interdisciplinary.

We have used rigorous systematic review and qualitative analytical methods complemented by a survey to develop the taxonomy. Investigators often combine multiple techniques in any one 'placebo' (eg, create positive expectancy+detailed history+symptom monitoring) making it beyond the scope of this project to unpack the effectiveness of individual techniques. Procedures did not always fit neatly into single domains. For example, 'screen for treatment history' was used to select patients for studies of specific treatments (and was thus placed in the patient's beliefs and characteristics domain), but its clinical application involves selecting a treatment for a specific patient and so could be considered a treatment characteristic. Conceptually, we would expect interactions between these domains; for example, some procedures categorised in other domains probably operate through causal pathways involving patients' beliefs as proximal determinants of placebo effects.[60] We feel the benefits of having a hierarchical structure (modifiable as the taxonomy is refined with use) outweigh the difficulties inherent in classification. We could have used many published reviews of placebo studies in non-malignant pain to identify original studies to review. Selecting seven such reviews means not using others; thus, we might have missed original studies that would have suggested additional procedures. Surveying leading researchers and incorporating their suggestions somewhat mitigate this limitation. Our sample of researchers was intended to be purposive, in that we wanted to obtain the views of leading researchers in the field. By using multiple means of identifying such individuals internationally, we feel we have achieved this. The reviews that we selected as the source of our papers and the papers themselves are now somewhat old examples of the literature, and our choice to exclude papers published before 1983 was arguably somewhat arbitrary. Future work should review very recent papers and iteratively improve the taxonomy accordingly.

This review extends previous work by Di Blasi et al,[31] building on their five domains to systematically develop a detailed taxonomy. We provide a new overarching framework that avoids the controversial and limited distinction between pure and impure placebos[61 62] and integrates ideas from the rich clinical and experimental literatures on placebo effects in non-malignant pain. Many of the components we have identified are likely to be important in other placebo-responsive conditions including depression,[63] IBS[64 65] and insomnia.[66] This taxonomy can guide two important and related applied research agendas: (1) to understand the components of placebo effects in clinical settings[46 67–70] and (2) to ethically harness evidence-based placebo effects to benefit patients.[14 15 71] We hope future studies might draw on the taxonomy to fully describe their methods and develop new applications, thus facilitating future systematic reviews and the development of a systematic and theory driven cumulative evidence base in this complex field.

The taxonomy identifies and classifies procedures that may contribute to placebo effects in clinical trials and experiments, providing an overarching framework for individual components. However, we do not suggest that every technique in this taxonomy will produce a placebo effect in every patient and we do not know from this project which techniques are more effective or how they might be combined to form ethically acceptable and effective complex interventions. This taxonomy provides the first attempt at a necessary conceptual tool to facilitate future research on these questions. For example, systematic reviews could use the taxonomy to code procedures in original studies, using this information in meta-regression analysis to examine the contribution of different procedures to placebo effects.[72] New clinical trials and experiments could extend existing work by systematically examining and comparing the effects and ethical acceptability of different procedures in the taxonomy, building a cumulative evidence base that has real pragmatic applicability to clinical practice. Some of the suggested clinical applications have been investigated more extensively in other literatures, in particular doctor–patient communication and the healthcare environment. This emphasises the need for a multidisciplinary approach to the translation of placebo research into practice. One fruitful way forward would be to draw on placebo theories to develop and test more mechanistic models of complex interventions intended to alter the context of healthcare encounters.

Placebo recipients in clinical trials and experiments are exposed to a large number and variety of procedures, many of which might contribute to placebo effects. Researchers seeking to develop a translational science of placebo effects are thus faced with myriad possibilities. We have systematically identified and defined these procedures, classified them into five domains and suggested possible clinical applications. The resulting taxonomy is presented as a preliminary but detailed and systematic guide for future research, which should in turn further refine the taxonomy. Ultimately, we hope to better conceptualise investigations of clinical applications of placebo effects in order to maximise opportunities for patient benefit.

**Acknowledgements** We acknowledge the contribution of all of the researchers who shared their views in the survey, including Przemyslaw Babel, PhD; Luana Colloca, MD, PhD; Professor Michael Doherty; Vanda Faria, PhD; Professor Magne Arve Flaten, PhD; Sarah Goldingay, PhD; John Hughes, PhD; Professor Dr Robert Juette; Irving Kirsch, PhD; Karin Meissner, PD Dr. med. Habil.; Daniel E Moerman, PhD; Meike Shedden Mora, PhD; Donald D Price, PhD and Professor Dr Harald Walach. We thank Professor Ted Kaptchuk for comments on an earlier draft of this manuscript.

**Contributors** FLB designed and led the study, drafted the manuscript and is guarantor. FLB, GL, AWAG, HE and PL secured funding for the project. FLB designed the study with input and revisions from GL, BC, AWAG, HE and PL. BC led data collection and analysis with additional data collection and analysis by MH and DS. All authors contributed to data interpretation. FLB drafted the manuscript, and all authors revised it for important intellectual content. All authors had full access to all of the data in the study and can take responsibility for the integrity of the data and the accuracy of the data analysis.

**Funding** The project "Creating a Taxonomy to Harness the Placebo effect in UK primary care" was funded by the National Institute of Health Research (NIHR) School for Primary Care Research (SPCR) (project number 161). This paper presents independent research funded by the NIHR. Additional funding for BC was provided by Solent NHS Trust.

**Disclaimer** The views expressed are those of the author(s) and not necessarily those of the National Health Service (NHS), the NIHR or the Department of Health.

**Competing interests** None declared.

**Provenance and peer review** Not commissioned; externally peer reviewed.

**Data sharing statement** Data underpinning this paper are available on request from the corresponding author.

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
