## [Reviewer comments · BMJ Open]

ARTICLE DETAILS

TITLE (PROVISIONAL)	What Techniques Might be Used to Harness Placebo Effects in Non-Malignant Pain? A Literature Review and Survey to Develop a Taxonomy.
AUTHORS	Bishop, Felicity; Coghlan, Beverly; Geraghty, Adam; Everitt, Hazel; Little, Paul; Holmes, Michelle; Seretis, Dionysis; Lewith, George

VERSION 1 - REVIEW

REVIEWER	Christopher Dowrick University of Liverpool, UK
REVIEW RETURNED	30-Dec-2016

GENERAL COMMENTS	This paper makes an important contribution to the field of placebo research by offering, for the first time, a comprehensive taxonomy of techniques which have the capacity to enhance placebo in pain related conditions. It is the most useful generic contribution to the field since the 2001 Di Blasi Lancet paper. The authors have undertaken a comprehensive and rigorous review of existing studies, coupled with a validating exercise involving international experts in this field, in order to propose 30 specific, discrete placebo-enhancing techniques. They wisely eschew the unhelpful distinction between pure and impure placebos. They also, interestingly, indicate wide variation in the extent to which these techniques have been used (or at least systematically described) in previous studies. This taxonomy will provide fertile ground for future placebo research, within and beyond the area of non-malignant pain. I have a few minor comments and queries: 1. The 1983 cut-off, on socio-cultural grounds is arbitrary and needs further justification. Also, variation may be not only diachronic but synchronic, related to differing socio-cultural beliefs across the world. What proportion of these studies were carried out in Western Europe and Anglophone countries?2. The decision to exclude psychotherapies, on grounds that it is difficult to disentangle active ingredients from effect if meaning, has validity in relation to the more Rogerian or analytical types of therapy, but is less relevant to highly specified therapies such as CBT or PST. The authors could usefully clarify their definitions here.3. With regard to the sections on Patient-Practitioner interactions, it is valuable to see the recommendation that 'the patient is seen by a practitioner whose views/values are congruent with the patient's'.
---

	Did the authors find any specific evidence that compatibility on socio-demographic dimensions such as age, gender, ethnicity or class might offer additional placebo benefits?
--	--

REVIEWER	Alexander Scott University of British Columbia, Canada
REVIEW RETURNED	04-Jan-2017

GENERAL COMMENTS	This paper has developed a scheme to classify plausible sources of placebo effects in the study of non-malignant pain. A systematic review with duplicate reviewers was carried out, using a predetermined set of existing review articles, as the material from which the taxonomy was developed. Following the review, a draft was created and sent out to experts in the field, who are listed in the acknowledgements – this resulted in the addition of several new items not found in the original articles. The taxonomy had kappa values of 0.93 when tested by two researchers involved in the study. The manuscript reads very well, has been rigorously conducted, and has the potential to stimulate a more coherent body of interdisciplinary work aimed at assessing the magnitude and clinical utility of placebo effects, either as primary or adjunct clinical approaches. The strengths of the manuscript include the large body of reviewed studies, the apparently high reliability of the classification system, and the involvement of numerous experts in its development. The comments include some suggestions to further strengthen the manuscript, including some additional validation. Abstract, line 1. Statement “placebo effects can be large and clinically meaningful”. The authors need to cite data to support the use of the word “large,” or just delete and state “clinically meaningful.” E.g. in medical research, Cohen’s effect sizes are commonly known and referred to, so data demonstrating an effect size of 0.8 or more would be helpful to make this point. Abstract, line 17. The statement “We systematically analysed methods used to elicit placebo effects in 169 clinical and laboratory-based studies” is problematic, as it is stated later on that what was actually done was to analyze methods which were considered to plausibly harness the placebo effect. This is important, as some of the “placebo” methods identified probably relate more to experimental design issues attempting to limit variability, e.g. “Ensure patient is cared for by the same doctor.” It is possible that procedures were included in the original studies due to design issues, but have been flagged by this review as plausibly harnessing the placebo effect. I think that’s not a problem, given the intent of the paper, as long as the statement above is amended to “We systematically analysed methods which could plausibly be used to elicit placebo effects in 169 clinical and laboratory-based studies” Page 4, line 6 – statement – “There is compelling evidence that factors other than the active ingredients of treatment can have substantial effects on symptoms, particularly non-malignant pain 1 2.” Neither of these references directly supports this statement. #1 detected placebo effect sizes that were defined by the authors as moderate Cohen’s effect sizes, and reference 2 stated that the effect sizes varied widely among trials.
---

	Reference required for this statement – “the overall analgesic effect of an opioid derives not only from its specific pharmaceutical actions but also from the meaning that the patient experiences when consulting the doctor and taking the medicine.” Page 7, line 41 - between all the authors – should be among all the authors Oage 8, line 7 - Four procedures deemed very unlikely to produce placebo effects (e.g. Conveying a Neutral Therapeutic Message) were excluded, - this reader would have been interested to know the other three procedures which were excluded (can be relevant to readers interested in experimental design). Could they be listed here? Title of table 1 – the descriptions all pertain to the use of procedures in a research setting (some of which relate to experimental design) which could plausibly create placebo effects. Could the title of the table, and the reference to the table tin the text, be changed to reflect this? E.g. “Taxonomy of Procedures Which Can Plausibly Elicit Placebo Effects in Non-Malignant Pain” and title of column “Definition and use in research studies” This paper seems relevant to the discussion of patient-clinician interactions - The Influence of the Patient-Clinician Relationship on Healthcare Outcomes: A Systematic Review and Meta-Analysis of Randomized Controlled Trials John M. Kelley,1,3,* Gordon Kraft-Todd,1 Lidia Schapira,1,4 Joe Kossowsky,2,5,6 and Helen Riess1 Table 3 is problematic for me. Given that the list of studies to be included in the review was admittedly rather arbitrary, and more for taxonomizing purposes, the calculation of percentages could be seen as misleading. Could a reference be provided for this statement on page 21? “These procedures are thought to operate primarily through affective mechanisms such as reduced anxiety after telling one’s story and being listened to with empathy and acknowledged. “ Some of the procedures described as “eliciting placebo effects” seem to relate more to experimental design issues attempting to limit variability, e.g. Ensure patient is cared for by the same doctor.” Another example – “Read records before consultation.” How was it determined that this was included in clinical studies as a means to elicit a placebo response? The paper could be strengthened (prospectively validated) by pulling a random selection of placebo-controlled RCTs, (not used in the development of the taxonomy) and ensuring that the taxonomy can successfully and reliability categorize the methods used by individuals not involved in its original development.
--	--

REVIEWER	Steven Savvas National Ageing Research Institute, Australia
REVIEW RETURNED	10-Jan-2017

GENERAL COMMENTS	A well-written review, relevant and timely. I have a number of comments that may improve the manuscript: Major 1. item one of the taxomony in Table 2 (select participants based on treatment Hx) seems out of place with other items. I wonder if the wording is wrong? Maybe 'tailor treatment based on prior treatment history' is more to the point. pg 16 gives an example for this item, which indicates that tailoring treatment is the mechanism. If this item encompasses more than just this though, then a different example would be more insightful. The item is again mentioned on pg 22 2nd paragraph, but provides no further insight in how this item works? Basically, when would i use this item? Minor 2. first paragraph. the authors used the phrase 'meaning that a person experiences', which seems clumsy. is there not a better word? Such as the meaning derived by the person etc? 3. studies were excluded pre-1983. The rationale seems poor for why they were excluded (socio-culturo), especially as a related item is one of the 30 items in the taxomony. Maybe better to just say that only studies post 1983 were included.
---

REVIEWER	Horing, Bjoern Institute of Systems Neuroscience University Medical Center Hamburg-Eppendorf
REVIEW RETURNED	06-Feb-2017

GENERAL COMMENTS	The authors provide a framework of techniques applicable to induce or enhance placebo effects in clinical research and practice, focussing on pain as primary outcome. The framework is obtained by analyzing (with various qualitative methods) a number of primary articles compiled from relevant reviews. Overall, the article is well written and a welcome step in providing a systematic in which methods and results can be contextualized. It is less to be understood as toolbox and more as an overview of possible methods. The authors clearly state its limitations and introduce it as a tool which other researchers can utilize to refine and expand it. -- For ease of correction, I am referring to the pages of the manuscript proper (not the compiled PDF's pages), but the lines provided by the PDF. -- Major issues p. 4, Intro: There have been numerous previous discussions of translational issues of placebo research and translations to clinical practice - it would be suitable to reference some of these attempts in the introduction. For example, see Enck et al., Nature Rev Drug Disc 2013. The fact that the work explicitly "extends previous work by Di Blasi et al." (p 23, line 15) at systematization, but this is brought up only in the discussion - I suggest referencing this work in the intro, as well. For an unstructured but more concrete early attempt at translation, see Walach & Jonas, J Compl Alt Med 2004. -- Minor content issues p. 2, l 21: Maybe elaborate with a half-sentence the goal and content
---

of the survey, i.e., the fact that the preliminary systematic was presented to experts in the field who were asked for opinions and additions.

p. 3, line 18f: I would recommend that the authors add a statement to the tune that their work is not intended to be an exhaustive compilation of state-of-the-art methods used in placebo research, but a first step towards this goal, or a framework for such an endeavor; the limitation concerning recency and coverage could be combined. This is more of a suggestion than an actual shortcoming.

p. 4, line 8: While I am not personally enthusiastic of the placebo effect's conceptualization as "meaning response", it is the authors prerogative to use it. However, I caution that this definition is somewhat insular and, at least to my observation, rarely employed in dedicated placebo research, much less in clinical research. The authors may want to also point to a more conventional definition in the framework of expectancy (e.g. Colloca & Miller, *Philos Trans R Soc Lond B Biol Sci* 2011), which is methodologically closer to the actually observed behavior (i.e., whether expectancy is generated due to some "meaning" the person explicitly experiences, or due to implicit conditioned responses, has no bearing on the placebo effect).

p. 4, line 36: I caution that the utilization of "open placebos" is faced with several inconsistencies, which do not afford an ethical free pass. For example, to my knowledge, the concrete methods used during open placebo application include strong suggestions of efficacy of these placebos; these suggestions however are truthful only insofar they are based on empirical research where placebos are a) given in the framework of a clinical trial, where some probability exists that the participant actually receive active medication, or b) in dedicated placebo research, where the placebo is accompanied with deceptive instructions of efficacy. In both cases, expectancies other than "this has no active component and hence will not work" are generated. It is therefore still deceptive to assert that "inert treatment works", when only a select few studies have investigated such "zero expectancy"-control groups, often with less efficacy than the compared "above zero expectancy" group. This longish exposition is to say that the cited preliminary evidence, limited as it is to subjective endpoints, comes with a large caveat concerning translational into clinical practice. I would at least recommend a pointer to pertinent ethical considerations (e.g. the introspective Blease et al., *Bioethics* 2016; Groll, *J Appl Philos* 2011; issue 366(1572) of *Philos Trans R Soc Lond B Biol Sci*). I appreciate that the authors point out that they do not suggest ethical viability later on.

p. 6, line 44f: It is relevant and interesting to point to the sociocultural context as potential determinant of particular procedures' efficacy. That said, the selection of the year 1983 (leading to the odd time frame of 34 years to be covered in the manuscript) seems arbitrary. While this would be fair enough (most articles do not even address this limitation, which applies to contemporary cultural distance as well) - is there a rationale for selecting this particular time frame?

p. 10, line 5: Conceptually, the contents of this table go beyond "eliciting placebo effects". For example, point 3 (reduce negative expectancy) usually does not apply to the target symptoms of the intervention (in this case, pain), but to adverse side effects arising alongside the target symptom. That said, it point 3 may be the only aspect where no primary effect is being reinforced, so a one-fits-all table header may be asking too much.

p. 11, line 18: It may be a good idea to add "salient" to point 20, since only salient side effects will have any behavioral consequence.

For example, it has no bearing if an active placebo reduces some endocrine parameter, if the patient has no means of perceiving this alteration.

p. 11, line 16: Is the use of point 19, "Ineffective substances", not subsumed by points 16 and 20?

p. 12, line 31: It is unclear to me what is meant by "dispelling any misconceptions". I am presuming the authors mean that where cultural conceptions include wrong assumptions about physiology or pathology, these should be addressed. However, this would go against the grain of many practices in so-called alternative or integrative medicine, which may, for example, generate sizeable placebo effects solely by merit of some variant of vitalism, which is a "wrong" assumption by scientific standards of truth. I appreciate that this is not the platform to address such issues, but the phrase could be re-assessed.

p. 15, Table 3: I found this table to be only of limited value, since the selection of reviews and hence original articles was arbitrary in nature, and one-sixth of the procedures were suggested by expert reviewers and not derived from the corpus in the first place.

Furthermore, the authors themselves state that the procedures were not assessed in terms of efficacy and ethical applicability. Therefore, in my opinion, the numbers presented here are inconsequential except as a rough estimate. If the authors wish to retain the table, they should contextualize it a bit more, or point out that the table may be useful for identifying underresearched areas/blind spots (I may have overlooked such a paragraph).

--

Minor formal issues

p. 2, line 37: "The" should be lower case.

p. 3, line 3: The authors use "clinical trials" here, whereas they elsewhere (e.g. in the abstract) refer to "clinical practice", later to "translational research". The presented systematic is not necessarily exclusive to either - still I recommend settling on a consistent definition of where and how it could be applied.

p. 4, line 5: Here and elsewhere, I recommend using the term "component" instead of "ingredient", which has a more corporeal connotation.

p. 8, line 14 and line 45: The authors are using both "procedures" and "items" to refer to the units of their taxonomy. I suggest settling on and sticking to a single nomenclature to avoid confusion, (even at the cost of some repetitiveness. The inclusion of three categories (domains, procedures, applications), while intuitive, is already sufficiently complex for a casual reader. If the authors wish to retain both "procedures" and "items", they may want to stick to one or the other in the larger subsections of the paper, e.g. only use "items" in the results section.

p. 10, line 8 and 11 (and others): I would recommend settling on either patient/practitioner or participant/experimenter as subjects or dyad to which the presented taxonomy is applied. The authors could include a sentence in the introduction that it may be applied to both, but from there on use only one or the other, for clarity. The same applies to line 32, intervention/experiment - I suggest using either one or the other throughout the text. Alternatively, one could always use both experimental and clinical entities. Alternating between the two adds confusion.

p. 10, line 32: Consistency of white space before and after slashes /. The authors use at least three possibilities throughout the manuscript (x/y, x/ y, x / y).

p. 11, lines 20, 24: "Matched treatments" and "Maximised treatment

	procedures" should start in upper case, as does the rest of the table. p. 16, line 12f: Avoid repetition of "for example". p. 16, line 17: I suggest using "(as the latter group...)" instead of "(as this group...)", to avoid grammatical ambiguity. p. 22, lines 8, 13: "Procedure" and "technique" are largely synonymous. Maybe use "implementation" (of a procedure) instead? Hence, "It includes 60 theoretically plausible clinical implementations, subject to further...?" p. 24, line 28f: I suggest getting rid of the "measured" in "measured placebo effects". The first part of the second sentence is grammatically deficient, as it could be read as "We have systematically identified these procedures into five domains, and classified these procedures into five domains"... I suggest something to the tune of "We have systematically identified and defined these procedures. We have then classified them into five domains. Furthermore, we suggested clinical applications for each of the procedures." In general, I feel that the final paragraph appears hasty and does not add to the remainder of the discussion. Some redundancy is of course acceptable, but maybe the authors can revisit it somewhat. p. 25, Acknowledgements: These should be brought into a more uniform format or at least list all titles properly. For example, Irving Kirsch and Magne Flaten are certainly PhDs, as well; "Prof. Dr." is prepended for Robert Juette, hence it should read "PD Dr. med. Karin Meissner", as well; Professor is spelled out for Harald Walach but not for others.
--	--

VERSION 1 – AUTHOR RESPONSE

Reviewer: 1: Christopher Dowrick

Comment

This paper makes an important contribution to the field of placebo research by offering, for the first time, a comprehensive taxonomy of techniques which have the capacity to enhance placebo in pain related conditions. It is the most useful generic contribution to the field since the 2001 Di Blasi Lancet paper. The authors have undertaken a comprehensive and rigorous review of existing studies, coupled with a validating exercise involving international experts in this field, in order to propose 30 specific, discrete placebo-enhancing techniques. They wisely eschew the unhelpful distinction between pure and impure placebos. They also, interestingly, indicate wide variation in the extent to which these techniques have been used (or at least systematically described) in previous studies. This taxonomy will provide fertile ground for future placebo research, within and beyond the area of non-malignant pain.

Response

Thank you for these positive comments on our work.

Comment

1. The 1983 cut-off, on socio-cultural grounds is arbitrary and needs further justification. Also, variation may be not only diachronic but synchronic, related to differing socio-cultural beliefs across the world. What proportion of these studies were carried out in Western Europe and Anglophone countries?

Response

This is a very good point. We have further justified our choice of 1983 (lines 114-8) and reflected on this limitation in the Discussion section (lines 388-9). We have now extracted the countries in which the 169 studies were conducted; as might be expected, the majority (160/169) were conducted in Western European and Anglophone countries.

Comment

2. The decision to exclude psychotherapies, on grounds that it is difficult to disentangle active ingredients from effect if meaning, has validity in relation to the more Rogerian or analytical types of therapy, but is less relevant to highly specified therapies such as CBT or PST. The authors could usefully clarify their definitions here.

Response

We excluded any type of psychotherapeutic interventions (line 118). It would be interesting to explore in future whether other placebogenic techniques might be found by examining the psychotherapy literature, but this is beyond the scope of our current project.

Comment

3. With regard to the sections on Patient-Practitioner interactions, it is valuable to see the recommendation that 'the patient is seen by a practitioner whose views/values are congruent with the patient's'. Did the authors find any specific evidence that compatibility on socio-demographic dimensions such as age, gender, ethnicity or class might offer additional placebo benefits?

Response

We did not systematically search for such evidence; we retained it in the taxonomy because we feel it is a plausible contributor given the theoretical mechanisms of placebo effects and that it warrants further investigation.

Reviewer: 2: Alexander Scott

Comment

This paper has developed a scheme to classify plausible sources of placebo effects in the study of non-malignant pain. A systematic review with duplicate reviewers was carried out, using a predetermined set of existing review articles, as the material from which the taxonomy was developed. Following the review, a draft was created and sent out to experts in the field, who are listed in the acknowledgements – this resulted in the addition of several new items not found in the original articles. The taxonomy had kappa values of 0.93 when tested by two researchers involved in the study. The manuscript reads very well, has been rigorously conducted, and has the potential to stimulate a more coherent body of interdisciplinary work aimed at assessing the magnitude and clinical utility of placebo effects, either as primary or adjunct clinical approaches. The strengths of the manuscript include the large body of reviewed studies, the apparently high reliability of the classification system, and the involvement of numerous experts in its development. The comments include some suggestions to further strengthen the manuscript, including some additional validation.

Response

Thank you for these positive comments on our work.

Comment

Abstract, line 1. Statement “placebo effects can be large and clinically meaningful”. The authors need to cite data to support the use of the word “large,” or just delete and state “clinically meaningful.” E.g. in medical research, Cohen’s effect sizes are commonly known and referred to, so data demonstrating an effect size of 0.8 or more would be helpful to make this point.

Response

Amended as suggested.

Comment

Abstract, line 17. The statement “We systematically analysed methods used to elicit placebo effects in 169 clinical and laboratory-based studies” is problematic, as it is stated later on that what was actually done was to analyze methods which were considered to plausibly harness the placebo effect.

This is important, as some of the “placebo” methods identified probably relate more to experimental design issues attempting to limit variability, e.g. “Ensure patient is cared for by the same doctor.” It is possible that procedures were included in the original studies due to design issues, but have been flagged by this review as plausibly harnessing the placebo effect. I think that’s not a problem, given the intent of the paper, as long as the statement above is amended to “We systematically analysed methods which could plausibly be used to elicit placebo effects in 169 clinical and laboratory-based studies”

Response

Thank you for this astute suggestion which we have implemented.

Comment

Page 4, line 6 – statement – “There is compelling evidence that factors other than the active ingredients of treatment can have substantial effects on symptoms, particularly non-malignant pain 1 2.” Neither of these references directly supports this statement. #1 detected placebo effect sizes that were defined by the authors as moderate Cohen’s effect sizes, and reference 2 stated that the effect sizes varied widely among trials.

Response

We have amended the statement and added other references that better evidence it (lines 61-63).

Comment

Reference required for this statement – “the overall analgesic effect of an opioid derives not only from its specific pharmaceutical actions but also from the meaning that the patient experiences when consulting the doctor and taking the medicine.”

Response

Reference added (line 73).

Comment

Page 7, line 41 - between all the authors – should be among all the authors

Response

This sentence has been rewritten (lines 143-5).

Comment

Page 8, line 7 - Four procedures deemed very unlikely to produce placebo effects (e.g. Conveying a Neutral Therapeutic Message) were excluded, - this reader would have been interested to know the other three procedures which were excluded (can be relevant to readers interested in experimental design). Could they be listed here?

Response

We now list all 4 procedures excluded on grounds of plausibility (Methods/Data Extraction and Synthesis, lines 160-1).

Comment

Title of table 1 – the descriptions all pertain to the use of procedures in a research setting (some of which relate to experimental design) which could plausibly create placebo effects. Could the title of the table, and the reference to the table in the text, be changed to reflect this? E.g. “Taxonomy of Procedures Which Can Plausibly Elicit Placebo Effects in Non-Malignant Pain” and title of column “Definition and use in research studies”

Response

Amended as suggested (line 200).

Comment

This paper seems relevant to the discussion of patient-clinician interactions - The Influence of the Patient-Clinician Relationship on Healthcare Outcomes: A Systematic Review and Meta-Analysis of

Randomized Controlled Trials John M. Kelley,1,3,* Gordon Kraft-Todd,1 Lidia Schapira,1,4 Joe Kossowsky,2,5,6 and Helen Riess1

Response

While we agree this is a very useful review on the topic of patient-clinician interactions and their impact on health outcomes, we do not feel it is directly relevant to the results section of our manuscript as this section cannot also review evidence of efficacy of the different procedures across the domains.

Comment

Table 3 is problematic for me. Given that the list of studies to be included in the review was admittedly rather arbitrary, and more for taxonomizing purposes, the calculation of percentages could be seen as misleading.

Response

Reviewer 3 raised similar concerns and on his suggestion we have provided additional contextualisation and caveats regarding the interpretation of this table (lines 197-9).

Comment

Could a reference be provided for this statement on page 21? "These procedures are thought to operate primarily through affective mechanisms such as reduced anxiety after telling one's story and being listened to with empathy and acknowledged."

Response

Reference added to (indirectly) support this statement (line 337-8). As explained in the methods section, these judgements were made by the authors after detailed discussion in several meetings and the proposed mechanisms are acknowledged to overlap.

Comment

Some of the procedures described as "eliciting placebo effects" seem to relate more to experimental design issues attempting to limit variability, e.g. Ensure patient is cared for by the same doctor." Another example – "Read records before consultation." How was it determined that this was included in clinical studies as a means to elicit a placebo response?

Response

To be included in our taxonomy the original researchers did not have to claim to have used procedures with the express intention of eliciting placebo effects. Rather, as you have noted in your other comments, we took the decision that these procedures might plausibly impact placebo effects. We hope that revising Table 1 headings will help clarify this. We erred on the side of inclusivity in making these decisions.

Comment

The paper could be strengthened (prospectively validated) by pulling a random selection of placebo-controlled RCTs, (not used in the development of the taxonomy) and ensuring that the taxonomy can successfully and reliably categorize the methods used by individuals not involved in its original development.

Response

Unfortunately our funding has expired for this project. We agree that such an exercise would add value but we feel it is beyond the scope of what is possible at the moment. If the editor decides this is essential, we will need more time to undertake this additional analysis but we think that this will significantly delay the publication of this work. We intend to continue with developing our strategy for placebo research and will consider this to be an important element of our activity in future grants

Reviewer: 3 Steven Savvas

Comment

A well-written review, relevant and timely. I have a number of comments that may improve the manuscript:

Response

Thank you.

Comment

1. item one of the taxonomy in Table 2 (select participants based on treatment Hx) seems out of place with other items. I wonder if the wording is wrong? Maybe 'tailor treatment based on prior treatment history' is more to the point. pg 16 gives an example for this item, which indicates that tailoring treatment is the mechanism. If this item encompasses more than just this though, then a different example would be more insightful. The item is again mentioned on pg 22 2nd paragraph, but provides no further insight in how this item works? Basically, when would i use this item?

Response

The wording for this procedure is derived from the literature based primarily on clinical studies in which patients are selected for (or excluded from) a trial based on their treatment history. The potential clinical application detailed in Table 2 Column 2 clarifies how this could work in practice, i.e., prescribe a new type of intervention that a patient has not experienced before, in preference to prescribing a type of intervention that a patient has previously found ineffective.

Comment

2. first paragraph. the authors used the phrase 'meaning that a person experiences', which seems clumsy. is there not a better word? Such as the meaning derived by the person etc?

Response

Amended accordingly (line 64).

Comment

3. studies were excluded pre-1983. The rationale seems poor for why they were excluded (socio-cultural), especially as a related item is one of the 30 items in the taxonomy. Maybe better to just say that only studies post 1983 were included.

Response

We have added an additional rationale for this exclusion (lines 114-8) and reflected on the limitations of this approach in the discussion (lines 388-9).

Reviewer: 4 Bhorng Clemson

Comment

The authors provide a framework of techniques applicable to induce or enhance placebo effects in clinical research and practice, focussing on pain as primary outcome. The framework is obtained by analyzing (with various qualitative methods) a number of primary articles compiled from relevant reviews. Overall, the article is well written and a welcome step in providing a systematic in which methods and results can be contextualized. It is less to be understood as toolbox and more as an overview of possible methods. The authors clearly state its limitations and introduce it as a tool which other researchers can utilize to refine and expand it.

Response

Thank you for these positive comments on our work.

Comment

p. 4, Intro: There have been numerous previous discussions of translational issues of placebo research and translations to clinical practice - it would be suitable to reference some of these attempts in the introduction. For example, see Enck et al., Nature Rev Drug Disc 2013. The fact that the work

explicitly "extends previous work by Di Blasi et al." (p 23, line 15) at systematization, but this is brought up only in the discussion - I suggest referencing this work in the intro, as well. For an unstructured but more concrete early attempt at translation, see Walach & Jonas, *J Compl Alt Med* 2004.

Response

We have added reference to others' work on translational issues in placebo studies (Introduction, lines 79-82).

Comment

p. 2, l 21: Maybe elaborate with a half-sentence the goal and content of the survey, i.e., the fact that the preliminary systematic was presented to experts in the field who were asked for opinions and additions.

Response

Good suggestion, amended accordingly (lines 30-1).

Comment

p. 3, line 18f: I would recommend that the authors add a statement to the tune that their work is not intended to be an exhaustive compilation of state-of-the-art methods used in placebo research, but a first step towards this goal, or a framework for such an endeavor; the limitation concerning recency and coverage could be combined. This is more of a suggestion than an actual shortcoming.

Response

Good suggestion, amended accordingly (lines 56-8).

Comment

p. 4, line 8: While I am not personally enthusiastic of the placebo effect's conceptualization as "meaning response", it is the authors prerogative to use it. However, I caution that this definition is somewhat insular and, at least to my observation, rarely employed in dedicated placebo research, much less in clinical research. The authors may want to also point to a more conventional definition in the framework of expectancy (e.g. Colloca & Miller, *Philos Trans R Soc Lond B Biol Sci* 2011), which is methodologically closer to the actually observed behavior (i.e., whether expectancy is generated due to some "meaning" the person explicitly experiences, or due to implicit conditioned responses, has no bearing on the placebo effect).

Response

Added reference to expectations (line 65).

Comment

p. 4, line 36: I caution that the utilization of "open placebos" is faced with several inconsistencies, which do not afford an ethical free pass. For example, to my knowledge, the concrete methods used during open placebo application include strong suggestions of efficacy of these placebos; these suggestions however are truthful only insofar they are based on empirical research where placebos are a) given in the framework of a clinical trial, where some probability exists that the participant actually receive active medication, or b) in dedicated placebo research, where the placebo is accompanied with deceptive instructions of efficacy. In both cases, expectancies other than "this has no active component and hence will not work" are generated. It is therefore still deceptive to assert that "inert treatment works", when only a select few studies have investigated such "zero expectancy"-control groups, often with less efficacy than the compared "above zero expectancy" group. This longish exposition is to say that the cited preliminary evidence, limited as it is to subjective endpoints, comes with a large caveat concerning translational into clinical practice. I would at least recommend a pointer to pertinent ethical considerations (e.g. the introspective Blease et al., *Bioethics* 2016; Groll, *J Appl Philos* 2011; issue 366(1572) of *Philos Trans R Soc Lond B Biol Sci*). I appreciate that the authors point out that they do not suggest ethical viability later on.

Response

We have now made reference to the ethical debates on open-label placebos (line 77).

Comment

p. 6, line 44f: It is relevant and interesting to point to the sociocultural context as potential determinant of particular procedures' efficacy. That said, the selection of the year 1983 (leading to the odd time frame of 34 years to be covered in the manuscript) seems arbitrary. While this would be fair enough (most articles do not even address this limitation, which applies to contemporary cultural distance as well) - is there a rationale for selecting this particular time frame?

Response

We have added an additional rationale for this exclusion (lines 114-8) and reflected on the limitations of this approach in the discussion (lines 388-9).

Comment

p. 10, line 5: Conceptually, the contents of this table go beyond "eliciting placebo effects". For example, point 3 (reduce negative expectancy) usually does not apply to the target symptoms of the intervention (in this case, pain), but to adverse side effects arising alongside the target symptom. That said, it point 3 may be the only aspect where no primary effect is being reinforced, so a one-fits-all table header may be asking too much.

Response

We have amended table 1 title and headers based on reviewer 2's suggestions. We have tried but cannot devise a table heading that also clearly encapsulates item 3 while also remaining clear and precise. We would welcome further suggestions from the editor and reviewer

Comment

p. 11, line 18: It may be a good idea to add "salient" to point 20, since only salient side effects will have any behavioral consequence. For example, it has no bearing if an active placebo reduces some endocrine parameter, if the patient has no means of perceiving this alteration.

Response

Good suggestion, amended accordingly (p12, item 20 in table).

Comment

p. 11, line 16: Is the use of point 19, "Ineffective substances", not subsumed by points 16 and 20?

Response

While we agree the clinical implications would be similar, the usage in the literature is different across these items. Item 19 ineffective substances includes products not indicated, which are conceptually different to sham interventions (designed and intended by triallists to appear as the genuine intervention).

Comment

p. 12, line 31: It is unclear to me what is meant by "dispelling any misconceptions". I am presuming the authors mean that where cultural conceptions include wrong assumptions about physiology or pathology, these should be addressed. However, this would go against the grain of many practices in so-called alternative or integrative medicine, which may, for example, generate sizeable placebo effects solely by merit of some variant of vitalism, which is a "wrong" assumption by scientific standards of truth. I appreciate that this is not the platform to address such issues, but the phrase could be re-assessed.

Response

Rephrased to specify: Elicit patients' culturally embedded treatment and illness beliefs, preferences and expectations, dispelling any potentially harmful misconceptions (p14, table 2 item 5).

Comment

p. 15, Table 3: I found this table to be only of limited value, since the selection of reviews and hence

original articles was arbitrary in nature, and one-sixth of the procedures were suggested by expert reviewers and not derived from the corpus in the first place. Furthermore, the authors themselves state that the procedures were not assessed in terms of efficacy and ethical applicability. Therefore, in my opinion, the numbers presented here are inconsequential except as a rough estimate. If the authors wish to retain the table, they should contextualize it a bit more, or point out that the table may be useful for identifying underresearched areas/blind spots (I may have overlooked such a paragraph).

Response

We have amended our sentence about table 3 to clarify its purpose: Table 3 shows the frequency of use of each procedure in clinical and experimental studies, and is intended as both an approximate guide to whether the procedures derived primarily from one or other literature and as a means to highlight those procedures that are very common and very rare in the literature (lines 196-9).

Comment

p. 2, line 37: "The" should be lower case.

Response

Corrected (line 37)

Comment

p. 3, line 3: The authors use "clinical trials" here, whereas they elsewhere (e.g. in the abstract) refer to "clinical practice", later to "translational research". The presented systematic is not necessarily exclusive to either - still I recommend settling on a consistent definition of where and how it could be applied.

Response

Revised for consistency of use throughout; we retain reference to clinical practice and translational research as we see these as slightly different ways of developing and/or applying our taxonomy.

Comment

p. 4, line 5: Here and elsewhere, I recommend using the term "component" instead of "ingredient", which has a more corporeal connotation.

Response

Good suggestion, amended accordingly throughout.

Comment

p. 8, line 14 and line 45: The authors are using both "procedures" and "items" to refer to the units of their taxonomy. I suggest settling on and sticking to a single nomenclature to avoid confusion, (even at the cost of some repetitiveness. The inclusion of three categories (domains, procedures, applications), while intuitive, is already sufficiently complex for a casual reader. If the authors wish to retain both "procedures" and "items", they may want to stick to one or the other in the larger subsections of the paper, e.g. only use "items" in the results section.

Response

Good suggestion, amended accordingly throughout.

Comment

p. 10, line 8 and 11 (and others): I would recommend settling on either patient/practitioner or participant/experimenter as subjects or dyad to which the presented taxonomy is applied. The authors could include a sentence in the introduction that it may be applied to both, but from there on use only one or the other, for clarity. The same applies to line 32, intervention/experiment - I suggest using either one or the other throughout the text. Alternatively, one could always use both experimental and clinical entities. Alternating between the two adds confusion.

Response

We have chosen to use clinically-oriented terminology and have explained this in Methods/Validating the Taxonomy (lines 182-6).

Comment

p. 10, line 32: Consistency of white space before and after slashes /. The authors use at least three possibilities throughout the manuscript (x/y, x/ y, x / y).

Response

Reviewed and corrected throughout.

Comment

p. 11, lines 20, 24: "Matched treatments" and "Maximised treatment procedures" should start in upper case, as does the rest of the table.

Response

Corrected

Comment

p. 16, line 12f: Avoid repetition of "for example".

Response

Corrected

Comment

p. 16, line 17: I suggest using "(as the latter group...)" instead of "(as this group...)", to avoid grammatical ambiguity.

Response

Corrected

Comment

p. 22, lines 8, 13: "Procedure" and "technique" are largely synonymous. Maybe use "implementation" (of a procedure) instead? Hence, "It includes 60 theoretically plausible clinical implementations, subject to further...?"

Response

Corrected

Comment

p. 24, line 28f: I suggest getting rid of the "measured" in "measured placebo effects". The first part of the second sentence is grammatically deficient, as it could be read as "We have systematically identified these procedures into five domains, and classified these procedures into five domains"... I suggest something to the tune of "We have systematically identified and defined these procedures. We have then classified them into five domains. Furthermore, we suggested clinical applications for each of the procedures." In general, I feel that the final paragraph appears hasty and does not add to the remainder of the discussion. Some redundancy is of course acceptable, but maybe the authors can revisit it somewhat.

Response

We have revisited and revised this final paragraph (lines 421-429).

Comment

p. 25, Acknowledgements: These should be brought into a more uniform format or at least list all titles properly. For example, Irving Kirsch and Magne Flaten are certainly PhDs, as well; "Prof. Dr." is prepended for Robert Juette, hence it should read "PD Dr. med. Karin Meissner", as well; Professor is spelled out for Harald Walach but not for others.

Response

We took names and titles directly from details supplied by individuals during our survey. We have now

reviewed and amended these details to ensure consistency while retaining individuals' preferences and cultural differences in presentation (lines 441-446).

VERSION 2 – REVIEW

REVIEWER	Christopher Dowrick University of Liverpool, UK
REVIEW RETURNED	16-Mar-2017

GENERAL COMMENTS	The authors have satisfactorily addressed my concerns
---

REVIEWER	Alexander Scott University of British Columbia, Canada
REVIEW RETURNED	20-Mar-2017

GENERAL COMMENTS	Thank you for the thorough response to this reviewer's concerns and suggestions.
--

REVIEWER	Steven Savvas National Ageing Research Institute, Melbourne, Australia.
REVIEW RETURNED	31-Mar-2017

GENERAL COMMENTS	No further comments.
----------------------

REVIEWER	Bjoern Horing Institute of Systems Neuroscience University Medical Center Hamburg-Eppendorf Germany
REVIEW RETURNED	31-Mar-2017

GENERAL COMMENTS	With the revision of their manuscript entitled "What techniques might be used to harness placebo effects in non-malignant pain? A literature review and survey to develop a taxonomy", the authors have addressed all pertinent issues from my side. A few minor quibbles remain, which I would leave at the authors' discretion to address. -- For ease of correction, I am referring to consecutively numbered lines of the manuscript proper, not the compiled PDF's. -- Minor content issues line 64: I am not sure that "learning" is the correct psychological domain to locate "expectation" in. I suggest circumventing such semantic concerns by simply omitting "From a learning perspective", and maybe elaborate the term "expectation" a little. For example, "Expectations - for example, those generated by verbal suggestion, or previous experiences - play a key role in placebo effects." line 161: That the team is multidisciplinary seems pertinent, maybe even before this sentence. It may be worthwhile to consider adding a
--

	sentence elaborating HOW it is multidisciplinary (psychologists, clinical practitioners, etc.) -- Minor formal issues General: The text tends to be less concise than it could be, sometimes with a rushed feeling to it. While I fully understand the allure of verbosity, the reading experience could be improved (with a tolerable loss in precision) by omitting some words, especially adjectives. Some sentences could be split in two or more, maybe using semicolons. A few examples:  - line 25: Remove "applicable" - line 29: Replace "In a validation exercise, we..." with "For validation, we..." - line 82: Remove "techniques and" - line 91: Remove "observed" - line 92: Remove "in [the] future" - line 93: Replace "potential approaches to augmenting placebo enhancement of analgesia" with "options for augmenting placebo analgesia" - line 102: Remove "together" - line 124: Remove "into a piloted form" - line 147: Remove "critically" (it is a default assumption that research is done "critically"; that said, if "critical examination" is a technical term of some qualitative method I am not aware of, it is appropriate) - line 214: Remove "have learned to" - line 222: Remove "at all" - line 263: Remove "active" - &c. pp. line 45: use "identify" instead of "conceptualize"; a "factor" is a conceptual entity line 54: and the line 79: The sentence is a bit awkward; maybe use something like "Placebo researchers have pointed out the necessity of more translational research" etc.? That they have also "begun" it seems implicit. line 106: "for [a] list" line 111f.: Restructure; I would use a list, but if this is not desired, I would group it that "reported..., reported..." are following each other, not use "were published..., published...", and maybe use semicolons. Also I suggest splitting the exclusion section in several sentences. line 145: ". ." line 223/224: Check consistency in using plural/singular for research entities ("patients" versus "patient's"). line 232: The "in doing so" construct seems a bit off - maybe restructure? -- Please also note that my name (contrary to what my email-address may imply) is not Bhorng Clemson, but Bjoern Horing (or Björn, if the keyboard affords it). Clemson University is a former employer of mine.
--	--

VERSION 2 – AUTHOR RESPONSE

We would like to thank Reviewer 4 for his meticulous attention to detail in suggesting a number of discretionary revisions. We have addressed each of these as described below, and have submitted a

“tracked changes” version of the manuscript.

COMMENT: line 64: I am not sure that "learning" is the correct psychological domain to locate "expectation" in. I suggest circumventing such semantic concerns by simply omitting "From a learning perspective", and maybe elaborate the term "expectation" a little. For example, "Expectations - for example, those generated by verbal suggestion, or previous experiences - play a key role in placebo effects."

RESPONSE: Sentence amended as suggested. New sentence: "Expectations – which can be generated, for example, by verbal suggestion or previous experience - play a key role in placebo effects."

COMMENT: line 161: That the team is multidisciplinary seems pertinent, maybe even before this sentence. It may be worthwhile to consider adding a sentence elaborating HOW it is multidisciplinary (psychologists, clinical practitioners, etc.)

RESPONSE: Sentence amended as suggested. New sentence: "The multidisciplinary team of authors (including for example GPs, clinical and health psychologists, and complementary medicine specialists) then generated possible clinical applications of each of these 25 procedures."

We made the minor and discretionary suggested changes to use of language where they were in keeping with our preferred writing style and intended meaning.